# Nrf2 Pathway and Autophagy Crosstalk: New Insights into Therapeutic Strategies for Ischemic Cerebral Vascular Diseases

**DOI:** 10.3390/antiox11091747

**Published:** 2022-09-02

**Authors:** Yue Hu, Yumin Luo, Yangmin Zheng

**Affiliations:** 1Institute of Cerebrovascular Disease Research and Department of Neurology, Xuanwu Hospital of Capital Medical University, Beijing 100053, China; 2Beijing Municipal Geriatric Medical Research Center, Beijing 100053, China; 3Beijing Institute for Brain Disorders, Capital Medical University, Beijing 100053, China

**Keywords:** ischemic cerebral vascular diseases, oxidative stress, Nrf2, autophagy, crosstalk

## Abstract

Cerebrovascular disease is highly prevalent and has a complex etiology and variable pathophysiological activities. It thus poses a serious threat to human life and health. Currently, pathophysiological research on cerebrovascular diseases is gradually improving, and oxidative stress and autophagy have been identified as important pathophysiological activities that are gradually attracting increasing attention. Many studies have found some effects of oxidative stress and autophagy on cerebrovascular diseases, and studies on the crosstalk between the two in cerebrovascular diseases have made modest progress. However, further, more detailed studies are needed to determine the specific mechanisms. This review discusses nuclear factor erythroid 2-related factor 2 (Nrf2) molecules, which are closely associated with oxidative stress and autophagy, and the crosstalk between them, with the aim of providing clues for studying the two important pathophysiological changes and their crosstalk in cerebrovascular diseases as well as exploring new target treatments.

## 1. Introduction

Globally, stroke is the second leading cause of death among people over the age of 60 years, and also the leading cause of permanent disability, characterized by high morbidity, disability, mortality, and recurrence rates [1,2]. Generally, stroke occurs as two types: ischemic and hemorrhagic. Ischemic stroke is the main type of stroke and the third leading cause of disability worldwide [3]. Approximately 80–85% of stroke events occur due to thrombogenesis or embolism in the cerebral artery, which induces cerebral ischemia, resulting in an insufficient blood supply in the ischemic area, especially in the middle cerebral artery [4]. Owing to the complexity of the causes, the diversity of pathophysiological changes, and the instability of disease changes after the onset of cerebrovascular diseases, current treatment strategies are imperfect; thus, side effects often occur. Currently, recombinant tissue plasminogen activator (rtPA) thrombolysis and arterial mechanical thrombectomy are clinically applied for the treatment of ischemic stroke. However, there is a strict treatment time window (4.5 h after stroke onset) for thrombolysis, which makes rtPA thrombolysis suitable for only approximately 5% of stroke patients [5]. In addition, this therapy can induce serious side effects, such as blood-brain barrier (BBB) damage and hemorrhage transformation [6]. Therefore, it is necessary to explore new effective clinical treatments with less toxicity and fewer side effects.

Drugs with significant therapeutic effects on ischemic stroke have become a hot topic in current research, especially active ingredients extracted from natural sources, because of their few associated side effects [7]. However, studies have mainly targeted the pathophysiological changes of stroke patients. The glutamate excitatory toxicity increases due to the decreased blood supply in ischemic areas with an insufficient energy and oxygen supply after an ischemic stroke. With an increase in the excitatory toxicity, the intracellular concentrations of sodium and calcium ions, along with the extracellular concentrations of potassium ions increase. Intracellular calcium overload leads to an increased release of free radicals and NO, resulting in mitochondrial dysfunction and DNA damage, thus inducing various pathophysiological changes, such as oxidative stress, autophagy hyperactivation, inflammatory response, neuronal apoptosis, and vascular damage [8]. Various pathophysiological activities are activated successively or simultaneously, forming a complex network. Diverse pathways and molecular interactions complicate the pathophysiological activities associated with stroke. For instance, oxidative stress and autophagy play important roles in pathological changes.

Oxidative stress is caused by an imbalance in the oxidant and antioxidant activities, generating an excess of reactive oxygen species (ROS), which oxidize macromolecular substances, resulting in attendant cellular damage [9]. Oxidants participate in cell death/survival signaling pathways and mediate mitochondrial dysfunction, resulting in reduced neuronal cell survival and tissue damage after cerebral ischemia [10]. Multiple in vivo antioxidant mechanisms activate to prevent the damage caused by oxidative stress after its initiation, and the main resistance system involves antioxidant enzymes. These enzymes include the erythrocyte antioxidant enzymes superoxide dismutase (SOD), glutathione peroxidase (GSH-Px), and catalase (CAT) [11]. In addition, the expression of antioxidant genes increases, among which Nrf2 plays a key role as a transcription factor that initiates the expression of antioxidant genes. This review details the structure and Nrf2- involved in ischemic cerebrovascular diseases.

Autophagic hyperactivation, another important pathophysiological activity, also plays an important role in the development and evolution of cerebral ischemia. Under physiological conditions, moderate autophagy plays a neuroprotective role [12]. Microtubule-associated protein 1 light chain 3 (LC3-Ⅰ), located in the cytoplasm, hydrolyzes into microtubule-associated protein 2 light chain 3 (LC3-Ⅱ), which then binds to the membrane after autophagy activation [13]. Therefore, LC3-Ⅱ can be used as a localization index for autophagosomes in tissues, and the formation of autophagic inclusion body particles containing LC3-Ⅱ is a marker of autophagy activation. Studies have found that autophagic cell death after cerebral ischemia is accompanied by an increase in Beclin 1 (a marker protein for autophagy) and LC-3; therefore, it can be inferred that neuronal death after cerebral ischemia may be related to overactivated autophagy [14,15]. Hence, we can infer that the overactivation of autophagy after cerebral ischemia may aggravate brain injury. This review describes in detail the processand function of autophagy, as well as autophagy-involved in ischemic cerebrovascular diseases.

The activation of the Nrf2 signaling pathway and autophagy is involved in the pathophysiological activities of cerebrovascular diseases. Some studies have been conducted on their crosstalk in cerebrovascular diseases, but further, more in-depth studies are needed to determine the underlying mechanisms. This review aims to provide prompts for future research on cerebrovascular diseases by analyzing the crosstalk between these two pathophysiological activities.

## 2. Nrf2 Pathway

### 2.1. Structure of Nrf2

Nrf2 is a member of the Cap-n-collar (CNC) family of basic leucine zipper (bZIP) proteins and is an activator of β-globulin genes. Nrf2 is a 66 kDa protein containing 605 amino acid residues and seven highly conserved domains, the Nrf2-ECH homology (Neh) domains, in humans [16]. These seven domains are involved in regulating structural stability and transcriptional activity. The N-terminal domain of Nrf2 is the Neh2 domain, which is responsible for regulating the interactions between Nrf2 and Kelch-like-ECH-associated protein 1 (Keap1), a Nrf2 inhibitor, at nanomolar concentrations (K_D_ ~5 nM), as well as the stability of Nrf2, and the degradation of ubiquitin. The Neh2 domain interacts with Keap1 through two motifs: high-affinity ETGE (K_D_ ~5 nM) and low-affinity DLG (K_D_ ~1 μM). Both motifs are essential for Keap1 regulation [17,18,19]. The Neh1 domain contains a basic leucine zipper module that acts as a heterodimer with the transcription partner small Maf proteins (sMafs) and binds DNA, allowing Nrf2 to bind to the antioxidant response element (ARE) sequence. In addition, the Neh1 domain can bind to UbcM2, a ubiquitin ligase of E2, to regulate Nrf2 protein stability. Upon the release of Neh1 from Keap1, the nuclear localization signal necessary for the nuclear translocation of Nrf2 is exposed [20]. The C-terminal domain of Nrf2 is the Neh3 domain, which interacts with the transcription coactivator CHD6 (a chromo-ATPase/helicase DNA-binding protein) and is responsible for the transactivation of ARE-dependent genes after nuclear chromatin remodeling [21]. Neh3 is a trans-activator domain, and the deletion of the Neh3 domain causes Nrf2 to lose its ability to bind ARE for gene expression, maintaining intact dimers, DNA binding, and subcellular localization [21]. Neh4 and Neh5 are transcriptional activation domains that bind to coactivator cyclic adenosine monophosphate response element-binding proteins to activate transcription [22,23]. The Neh5 domain also regulates Nrf2 cellular localization, while Neh6 regulates the Keap1-dependent degradation of Nrf2 and represents a binding platform for the β-transducin repeat-containing protein (β-TrCP) [20]. β-TrCP is a substrate adaptor for the S-phase kinase-associated protein 1 (SKP1)–Cul1–RING-box protein (Rbx1)/Roc1 ubiquitin ligase complex The Neh6 domain negatively regulates Nrf2 through DSGIS and DSAPGS motifs. DSGIS motif increases the ability of β-TrCP to ubiquitinate Nrf2 and promotes its rapid conversion [24,25]. Neh7 binds to the nuclear receptor–retinoic X receptor alpha (RXRα), which inhibits Nrf2 transcription [26]. The structure of Nrf2 is shown in Figure 1A.

Keap1 is a substrate adaptor for Cul3-containing E3 ubiquitin ligase, which is the main intracellular regulatory factor and cytoplasmic inhibitory protein of Nrf2. Keap1 has five domains, including the N-terminal region (NTR), the Broad-complex, Tramtrack and Bric-à-Brac (BTB) domain, the intervening region (IVR), the Kelch/double glycine repeat (DGR) domain, and the C-terminal region (CTR). Each of these plays an important role in inhibiting Nrf2 activity. The BTB is a homodimerization domain that sequesters Nrf2 in the cytoplasm and represses Nrf2 transactivation [27]. Keap1 binds Cul3 through its N-terminal BTB and binds to the Neh2 domain of Nrf2 through the C-terminal Kelch domain, thus bridging Cul3 and Nrf2 to regulate the degradation of Nrf2 [28]. The IVR domain contains active cysteine residues that act as redox sensors, and C273A and C288A are known to be necessary for Keap1 to inhibit Nrf2 activation [29]. The DGR domain contains six Kelch repeats, which are related to the Neh2 domain of Nrf2 bound to Keap1, and associates with p62 [30]. The structure of Keap1 is shown in Figure 1B.

Keap1 is a part of the Cul3-containing E3 ubiquitin ligase and strongly regulates the activity of the transcription factor Nrf2 through targeted ubiquitination and proteasome-dependent degradation. In homeostasis, two Keap1 molecules interact with one Nrf2 molecule through its DLG and ETGE motifs, thus promoting the ubiquitination and degradation of Nrf2. In the Nrf2-Keap1 complex, DLG functions as a latch, and ETGE functions as a hinge. The “hinge and latch” model forms the stress sensing mechanism, wherein DLG with a low affinity with Nrf2 acts as a latch to turn the degradation of Nrf2 on or off via ubiquitination. When oxidative stress and other stress factors occur, DLG dissociates from Keap1, and Nrf2 separates from the Nrf2-Keap1 complex, translocating into the nucleus, interacting with transcription partner sMafs and binding to ARE to begin transcription activity. This is the classic regulation of Nrf2 transcription. Advances in research have found that Keap1 has a variety of stress receptors and inactivation modes, from oxidative stress and cell metabolism to autophagy disorder, and the regulation of Nrf2 activity from multiple cellular inputs. In mammals, the Nrf2-Keap1 system plays a pivotal role in resisting stress and maintaining homeostasis. The molecular regulatory mechanism of Nrf2 is shown in Figure 2.

### 2.2. Nrf2-Involved in Ischemic Cerebrovascular Diseases

Based on different types of animal models of cerebral ischemia, numerous studies have reported the dynamic regulation of Nrf2 after cerebral ischemia. The cerebral ischemia animal models used in these studies included permanent cerebral ischemia models (pMCAO), transient cerebral ischemia models (tMCAO) and global cerebral ischemia models. In most pMCAO and tMCAO animals, the expression of the Nrf2 protein, its downstream antioxidant target genes and its expressed target proteins (heme oxygenase1 (HO-1), NAD(P)H: Quinone oxidoreductase I (NQO1), etc.) were upregulated in different ischemic sites (cortex, hippocampus, etc.) and different ischemic cells (neurons, astrocytes, microglia, etc.) after cerebral ischemia [31,32,33,34,35]. However, a few studies did not find a statistically significant increase in Nrf2 expression [36], or even a decrease in *Nrf2* mRNA expression [37], thus indicating the need for further exploration. In addition, studies on global cerebral ischemia models seem to be controversial owing to the influence of many factors, such as animal background, age, difference in ischemic time, and observations at different ischemic sites [38].

Nrf2 has been found to play a protective role after cerebral ischemia in several *Nrf2* knockout animal models [39,40]. In several pMCAO experiments, severe cortical infarct volumes and neurological deficits were found in *Nrf2* knockout animals [41,42,43]. Therefore, studies on experimental animal models of cerebral ischemia revealed that Nrf2 is a potential target for intervention in ischemic stroke. Nrf2 is expected to stimulate cellular protective responses against harmful ischemic events. At present, several candidate Nrf2 inducers have been identified in *Nrf2*-knockdown cerebral ischemia animal models that may play a protective role in the brain. These include dimethyl fumarate (DMF), monomethyl fumarate, sulforaphane (SFN), and tert-butylhydroquinone [43,44,45]. However, some studies have shown distinct results regarding the role of Nrf2 in cerebral ischemia, warranting further investigation. There are multiple possible reasons for the contrary results: selection differences for Nrf2 antibodies, different types of Nrf2-related cells that play a role in neuroprotection, different *Nrf2* knockout lines, whether the expression of protective Nrf2-activated enzymes is sufficiently high, whether the Nrf2 activator can pass through the BBB and directly activate *Nrf2*, differences in detected ischemic brain tissue sites, and differences in detected ischemia time points and markers. Owing to these factors, overall differences may arise in the experimental results. Hence, further rigorous and in-depth studies are necessary to provide information for Nrf2-targeted interventions and realize their prospects for clinical transformation and application [38].

Ischemic brain tissues are damaged by increased aseptic inflammation and oxidative stress caused by ROS, which results from the resupply of blood flow after cerebral ischemia-reperfusion (I/R). ROS can cause membrane lipid peroxidation and destroy the membrane integrity, leading to protein degeneration, DNA damage, and cell death. Consequently, ROS-stimulated oxidative stress is an important pathological change associated with cerebral I/R injury. It has been found that Nrf2 confronts oxidative stress through multiple signaling pathways. Several studies have found that the Nrf2-Keap1-ARE signaling pathway plays an anti-oxidative stress role in cerebral I/R injury [46]. Nrf2 activation acts as a transcription factor against oxidative stress and regulates the expression of antioxidant enzymes that act as ROS scavengers and electrophilic neutralizers. These enzymes include catalase, superoxide dismutase, thioredoxin, thioredoxin reductase, glutathione peroxidase, glutathione reductase, sulfiredoxin, NQO1, HO-1, glutaredoxin, and glutathione *S*-transferase [47,48,49]. Moreover, Nrf2 interacts with the NF-κB signaling pathway to regulate redox homeostasis and cellular responses to stress and inflammation [50]. Multiple signals activate NF-κB by degrading the IκB proteasome and subsequently translocating to the nucleus to induce the expression of inflammatory factors and anti-oxidative stress genes, thereby regulating cellular redox homeostasis, stress, and inflammatory responses. Furthermore, multiple studies have found that excessive ROS activates Nrf2, regulates the transcription and expression of the *p62* gene, and participates in the protein degradation regulated by autophagy after cerebral ischemia [51]. Therefore, the ROS-Nrf2-p62 signaling pathway regulates oxidative stress and autophagy after cerebral ischemia and plays a protective role in the cells. Glycogen synthase kinase 3β (GSK-3β) is an inhibitor of Nrf2, which functions by inhibiting nuclear translocation and degrades Nrf2 by phosphorylation [24]. This process occurs in a Keap1-independent manner. However, GSK-3β is activated by hydrogen peroxide; therefore, the regulation of Nrf2 by GSK-3β may also occur through oxidative stress sensitivity. Recent studies have focused on modulating Nrf2 by GSK-3β to counter the altered oxidative environment of brain tissues after cerebral ischemia. However, previous studies in some animal models have shown the opposite results regarding the possibility of GSK-3β regulating Nrf2 and protecting against cerebral ischemia, while some existing studies have defects in their research methods. Therefore, whether GSK-3β inhibition can enhance Nrf2 activity and play a protective role after cerebral ischemia requires further exploration and confirmation [52].

The increased permeability of the BBB and impairment of neurological function will occur after an ischemic stroke. The Nrf2 activator SFN alleviates the blood–brain barrier injury and inhibits disease progression by activating Nrf2 and HO-1 in perivascular astrocytes around infarction, thus alleviating neurological impairment. Consequently, the activation of the Nrf2-HO-1 signaling pathway may play a neuroprotective role after cerebral ischemia [44]. Other studies have found that lycopene pretreatment of the bilateral common carotid artery occlusion (BCCAO) mouse model for 7 days can reduce neuronal apoptosis and improve neurological function scores by activating the Nrf2/HO-1 pathway [53]. Additionally, DMF induces the Nrf2/HO-1 pathway. Using a rat MCAO model, Yang et al. found that DMF significantly reduced the volume of cerebral infarction, alleviated cerebral edema, decreased cell death, and improved neurological defects [43]. In a study on the neuroprotective effect of andrographolide on stroke, Yang et al. found that a p38 mitogen-activated protein kinase (MAPK) inhibitor significantly inhibited Nrf2 phosphorylation and reduced the expression of the *Nrf2* downstream target gene, *HO-1*, suggesting that p38MAPK may be an upstream regulatory factor for Nrf2 activation. In summary, andrographolide plays a neuroprotective role by activating the p38MAPK/Nrf2/HO-1 signaling pathway [54].

The Nrf2 signaling pathway can not only combat oxidative stress, but also promotes neovascularization. Huang et al. found that under hypoxia, *Nrf2*-knockout brain microvascular endothelial cells can inhibit VEGF expression through the PI3K/AKT pathway, thereby inhibiting angiogenesis [55]. Data indicated that hypoxia induces a transient increase in Nrf2, and that Nrf2 plays an important role in cerebral microangiogenesis. In hypoxia-induced bEnd.3 cells, Nrf2 may regulate the proliferation and migration of vascular endothelial cells and the formation of tubular structures through the PI3K/AKT pathway [55]. In addition, Bai et al. found in a tMCAO rat model that epigallocatechin-3-gallate, the active ingredient in green tea, can promote angiogenesis and reduce oxidative stress by activating the MAPK/Nrf2/VEGF signaling pathway [56].

In addition to its critical role in ischemic stroke, Nrf2 is also involved in other ischemic cerebrovascular diseases. Endothelial cell injury is an early pathological change in cerebral small vessel disease, and oxidative stress is the key factor in this process. Increased Nrf2 expression can significantly improve endothelial cell injury by reducing oxidative stress [57]. Mixed dementia presents with mutually reinforcing neurodegeneration and vasculopathy, with all the features of Alzheimer’s disease (AD) and vascular dementia. The enhancement of cAMP and/or cGMP signaling has been found to exert beneficial effects via Nrf2 activation [58].

## 3. Autophagy

### 3.1. Process of Autophagy

Autophagy is an intracellular evolutionarily-conserved process; it is an important catabolic pathway that eliminates cytoplasmic components through lysosomal degradation and recirculates cellular materials, such as longevity proteins, organelles, and pathogens, to maintain cellular homeostasis. It is the main regulatory factor for cells and the body to adapt to various endogenous and exogenous stresses [59]. Three types of autophagic processes have been described in the literature: Microautophagy, chaperone-mediated autophagy, and macroautophagy [60]. Autophagy was previously considered a process involving the non-selective removal of cytoplasmic components during cell starvation [61]. However, with the development of research, autophagy has been found to play an important role in the selective removal of specific receptors. Consequently, autophagy is now divided into the following categories, according to the removed components: aggrephagy (protein aggregates), xenophagy (invading pathogens), mitophagy (mitochondria), pexophagy (peroxisomes), lipophagy (lipid droplets), and reticulophagy (endoplasmic reticulum [ER]) [62].

The most commonly described process of autophagy is macroautophagy [63]. Autophagy is a lysosome-dependent degradation pathway involving approximately 16–20 core conserved autophagy-associated proteins, which are encoded by autophagy-associated genes (ATGs). Autophagy consists of five stages: nucleation of the phagophore membrane, expansion, phagophore closure, fusion between autophagosomes and multivesicular endosomes or lysosomes, and the degradation of the autophagosome contents [64]. The initial stage of autophagy involves the formation of autophagosomes from phagophores. Autophagosome formation begins with the formation of protein kinase complexes including ULK1/2, ATG13, RB1CC1 (also known as FIP200 or ATG11), and ATG101. This complex can be activated by two types of activation signals, the first being the mechanistic target of rapamycin complex 1 (mTORC1) and AMP-activated protein kinase (AMPK) signals. A variety of nutrients, such as amino acids, growth factors, and stress factors, act on and activate mTORC1. mTORC1 inhibits ULK1/2 in the normal physiological state. However, in a state of starvation, this inhibition is suppressed, leading to the translocation of the initiation complex to the site of autophagosome formation. Additionally, its low-energy status activates ULK1/2 through AMPK. The second is the autophagic cargo activation signals; for instance, damaged mitochondria activate the complex by interacting directly with FIP200. In addition, autophagosomes are formed by AKT and EGFR, inducing autophagy through the phosphorylation of Beclin 1 independently of mTORC1. When this protein kinase complex is activated, ULK1/2 phosphorylates components of the class III phosphatidylinositol 3-kinase (PI3K) complex and generates phosphatidylinositol 3-phosphate (PI3P) on the autophagosomal precursor membrane to form an autophagosomal membrane. The PI3K complex contains ATG14, Beclin 1, VPS34, and VPS15. ATG9 vesicles are involved in this process. Subsequently, PI3P interacts with members of the WIPI protein family (WIPIs) and recruits ATG2A/B proteins. 

In the elongation phase, WIPI2 recruits ATG12-ATG5 and ATG16L, which promote the ATG8 protein family (LC3 and GABARAP subfamilies) to bind to phosphatidylethanolamine (PE). This process is mediated by two ubiquitin-like conjugation systems: (1) The conjugation of PE to cytoplasmic LC3-I to generate the lipidated form, LC3-II, which is facilitated by the protease ATG4B, the E1-like enzyme ATG7, and ATG3, whereby LC3-II is incorporated into the growing membrane. The lipidation of LC3-Ⅱ is crucial for the capture of autophagic cargo and stability of the inner membrane of autophagosomes. (2) The second conjugation system is mediated by ATG7 and the E2-like enzyme ATG10, resulting in an ATG5–ATG12 conjugate. The edges of autophagosomes are eventually sealed by the endosomal sorting complex required for transport (ESCRT). After its closure, autophagosomes and lysosomes fuse to form autophagolysosomes. Eventually, the low pH of the lysosome results in the degradation of the autophagosomal contents. The process of autophagy is illustrated in Figure 3.

### 3.2. Functions of Autophagy

Autophagy plays a critical role in the biological activities of most eukaryotes. In fact, it has been found to be involved in core metabolism, injury regulation, and cell death in eukaryotes. There are numerous reports on the functions of macroautophagy; the lysosomal degradation pathway of macroautophagy plays a crucial role in cell physiology, including adaptation to metabolic stress, removal of dangerous cargo (such as protein aggregates, damaged organelles, and intracellular pathogens), repair, and prevention of gene damage during development and differentiation. Autophagy has been shown to play protective roles in cancer, neurodegenerative diseases, infections, inflammatory diseases, aging, and insulin resistance in animal models [65]. The level of autophagy increases and cells change their nutritional and metabolic requirements to adapt to stress demands through protein catabolism under stress conditions. Sousa et al. found that pancreatic astrocytes regulate tumor metabolism through the autophagy of alanine as a source of tricarboxylic acid circulating carbon supplementation [66]. He et al. found that acute exercise enhances autophagy in the skeletal and cardiac muscles of fed mice, and autophagy induction may play a beneficial role in post-exercise metabolism [67]. Fernandez et al. found that increased autophagy reduces age-related renal and cardiac pathological changes and spontaneous tumor occurrence, reduces premature death and infertility, prevents premature aging, improves health, and extends the lifespan in mammals [68]. At the cellular level, organelles regulate cell metabolism, thus controlling cell death. Mitochondria can be regulated by autophagy, constantly adjusting their shape to adapt to changing biological energy requirements, or undergoing a lethal permeabilization process that triggers apoptosis [69]. Khaminets et al. proposed that selective reticulophagy is essential for the maintenance of mammalian cell homeostasis through the regulation of ER turnover [70]. Reduced cellular nutrient levels can activate autophagy, providing nutrients to maintain cellular homeostasis by degrading macromolecules [71]. However, in some special cases, autophagy may have deleterious effects through its pro-survival effect (e.g., promoting cancer development) [72] or, possibly, its cell death-promoting effects [73].

With the progress of research, it has been found that components of the autophagy apparatus can mediate non-autophagic functions [73], some of which are related to membrane biological functions. These include: (1) endocytosis, phagocytosis, and intracellular vesicular trafficking; (2) conventional and non-conventional secretion; and (3) cytokinesis. In addition, they can regulate membrane-unrelated functions, including: (1) inflammatory and immune responses; (2) cell death; (3) genomic stability; and (4) cell proliferation. Autophagic and non-autophagic functions regulated by autophagy-related proteins are summarized in Table 1.

### 3.3. Autophagy-Involved in Ischemic Cerebrovascular Diseases

Current studies on the effects of autophagy on human diseases are mainly carried out using animal models of autophagy knockout. Mutations in genes that regulate the expression of proteins involved in autophagy can impair autophagy, and different mutations can cause different diseases [123]. The abnormal regulation of autophagic activity can lead to enhanced or inhibited autophagy in different diseases, and even different pathophysiological processes of the same disease may have different effects.

After cerebral I/R, pathophysiological processes, such as mitochondrial dysfunction, oxidative stress, inflammatory response, calcium overload, and excitatory toxicity, can occur, resulting in cell damage and tissue destruction, thus inducing autophagy. On the one hand, recent studies have found that severe acute cerebral ischemia induces hyperactivated autophagy and promotes cell damage and death; on the other hand, chronic and mild hypoxia activates moderate autophagy, which plays a protective role by removing damaged proteins and tissues. Therefore, autophagy plays contrasting roles in cerebral ischemia. Autophagy is involved in cerebral ischemia through the following signaling pathways: (1) Autophagy is mediated by the mTOR signaling pathway, and studies have shown that mTOR negatively regulates autophagy. Hei et al. found that the inhibition of mTOR after cerebral ischemia could moderately induce autophagy and protect against cerebral ischemia in rats with acute hyperglycemia [124]. (2) Autophagy regulated by the MAPK signaling pathway is regulated by p38, extracellular regulated protein kinases (ERK), and c-Jun N-terminal kinase (JNK). Multiple studies have shown that several drugs can enhance the protective effect of autophagy after cerebral ischemia by activating ERK, and inhibiting JNK and p38MAPK [125,126]. p38 inhibitors can improve mitochondrial injury induced by ischemia, reduce the volume of cerebral infarction, and promote the recovery of neurological function by activating the cell survival signaling pathway [127]. (3) Autophagy is associated with cerebral ischemia through the hypoxia-inducible factor-1α (HIF-1α)-regulated signaling pathway—previous studies have found that HIF-1α is closely associated with neuronal death after cerebral ischemia. When HIF-1α is overexpressed, mitochondrial autophagy is activated and the mTOR signaling pathway is inhibited, thus improving neuronal survival. For instance, the transplantation of HIF-1α-overexpressing bone marrow mesenchymal stem cells into MCAO rats reduced the volume of cerebral infarction and improved the neurological function scores by inhibiting the mTOR signaling pathway and activating AMPK [128].

Autophagy has different effects on different cells after cerebral ischemia. Since neuronal cells are the most sensitive to ischemic injury, neurons are the first to raise autophagy after ischemic stroke. In recent years, studies on the effects of autophagy on neurons induced by cerebral ischemia have shown contradictory results. Some studies have found that autophagy has a protective effect against neuronal injury after cerebral ischemia [129,130], while others have found that autophagy inhibition can protect neurons from ischemic injury [131,132]. According to previous research, we speculate that this may be because of the cerebral ischemia models used in the experiment, and ischemia time and intensity differences caused by differences in autophagy activation; moderate autophagy can protect ischemic neurons, and the excessive activation of autophagy may aggravate neuronal injury after ischemia, but the mechanism is currently not completely clear. The dosage and mode of administration may affect the effect of the drug. The specificity of the drug is not strong, so autophagy may have unrelated effects and the instability of the variable degree of cerebral ischemia caused by the operator of the cerebral ischemia model results in data errors. It should be noted that the difference in the usage of neuroprotective agents in animal models and clinical trials is an important issue in the treatment of stroke. Microglia are resident immune cells in brain tissues. After cerebral ischemia, microglia are activated and promote autophagy. Yang et al. found that hypoxic treatment increased HIF-1α levels and promoted autophagic death in microglia cultured in vitro [133]. Chen et al. found that activated autophagy after ischemia-reperfusion regulates microglial apoptosis through the ROS-regulated Akt/mTOR signaling pathway [134]. Autophagy activated by microglia can regulate the phenotypic changes of microglia through the NF-κB pathway, which is beneficial to neuronal repairment after cerebral ischemia [135]. Astrocytes are the most abundant cell type in brain tissue and contribute to the formation of the BBB. After cerebral ischemia or in vitro cultured astrocytes suffered from oxygen and glucose deprivation (OGD), the autophagic activity of astrocytes was enhanced. Using autophagy inhibitors to dispose of astrocytes, OGD can reduce autophagic death by inhibiting autophagy. However, several studies have reported contradictory findings. Kasprowska et al. found that the results of short-term OGD treatment and long-term OGD treatment followed by treatment with an autophagy inhibitor were completely opposite for astrocytes cultured in vitro [136], suggesting that the effect of autophagy on astrocytes may be related to the time and intensity of ischemia in astrocytes. Furthermore, some studies have shown that autophagy has opposing effects on the BBB after cerebral ischemia [137,138]. On the one hand, the activation of autophagy after cerebral ischemia promotes the degradation of the blood–brain barrier protein, Claudin-5. On the other hand, the upregulation of autophagy can reduce the damage to the cerebral microvascular endothelial cells after cerebral I/R, thus reducing damage to the BBB [139]. Therefore, it is once again confirmed that autophagy is a double-edged sword in the context of cerebral ischemia.

Since autophagy plays a dual role in ischemic stroke, a combined treatment strategy can be adopted to target it: (1) maintain moderate autophagy before the onset of ischemic stroke; and (2) inhibit the overactivation of autophagy after ischemic stroke. It should be noted that autophagy can become over-activated at any time, and targeting a single molecule or signaling pathway will not be sufficient to exert a beneficial role. Therefore, the development of a broad range of autophagy regulators to achieve a holistic targeted intervention may be a potential therapeutic strategy [1].

In addition to playing an important role in ischemic stroke, autophagy is also involved in other ischemic cerebrovascular diseases. Cerebral autosomal-dominant arteriopathy with subcortical infarcts and leukoencephalopathy (CADASIL) is the most common form of hereditary brain small-vessel disease characterized by *NOTCH3* gene mutations. Patients with CADASIL were found to have *SQSTM1* gene mutations, with the damaged *NOTCH3* and *p62* genes encoding key proteins in the autophagic lysosomal pathway [140]. In addition, autophagic lysosomal pathway dysfunction was found in the vascular smooth muscle cells of patients with CADASIL [141]. Therefore, autophagy is an important component of the CADASIL pathology. Damaged autophagy impairs the clearance of dysfunctional mitochondria, causing a large amount of mitochondrial DNA to be released into the cytosol, inducing the activation of caspase-1 and the subsequent induction of IL-1β production. These responses thus cause a low-grade, sterile, long-term inflammatory reaction [142]. In addition, aging diabetic rats are prone to cerebral small-vessel diseases accompanied by cognitive impairment, which is closely related to Beclin1-regulated autophagy in the hippocampus [143]. The important pathological changes of vascular dementia are cerebral microvascular injury and BBB disruption. Autophagy plays a key role in maintaining vascular integrity, neuronal homeostasis, and the integrity of the BBB. After vascular dementia occurs, abnormal proteins, lipids, and plaques are deposited in the brain. Regulated autophagy can inhibit the pathological process and maintain normal vascular physiology [144]. Abnormally activated autophagy reduces the clearance of protein aggregates and participates in the occurrence of pathological events. Cerebral amyloid angiopathy (CAA) is an important cause of vascular dementia. The deposition of amyloid plaques in cerebral arteries is the main pathological event of cerebral amyloidosis. Abnormally activated autophagy promotes the formation of amyloid plaques in cerebral arteries, reduces the removal of protein aggregates, and increases the likelihood of vascular wall collapse [145].

## 4. Crosstalk between Nrf2 Pathway and Autophagy

At present, there are numerous studies on the crosstalk between autophagy and the Nrf2-Keap1 signaling pathway. An autophagy-deficient rat model has shown that the dysregulation of autophagy can cause the long-term activation of Nrf2, resulting in tissue injury and cancer. In addition, the absence of autophagy-related proteins ATG5, ATG7, and Beclin1 reduces autophagy activity; thus, p62 binding to Keap1 and aggregating in the cytoplasm, promotes the prolonged activation of Nrf2 [146,147,148,149]. Komatsu et al. conducted valuable mechanistic studies on the crosstalk between autophagy and Nrf2 in *ATG7*-deficient mice. ATG7 has mainly two functions: the first is to act as an ubiquitin-activating enzyme E1 during the initiation of autophagosome synthesis, and second, it plays an important role in the extension of phagophores [150]. The inhibition of the phagophore elongation process was observed in *ATG7*-deficient rat models, with the enrichment of protein aggregates p62 and Keap1 [151].

The functional association between autophagy dysfunction and the activation of the Nrf2 signaling pathway appears to be realized mostly through physical interactions between the autophagy adaptor p62 and Keap1. P62 is an autophagy adaptor protein that regulates the formation of protein aggregates that are cleared by autophagy. P62 acts as a ubiquitination target for the autophagic degradation of cargo receptors, binding to ubiquitinated protein aggregates, and their transport to autophagosomes. It can be upregulated under the action of various stress factors. Komatsu et al. found that *p62* gene ablation inhibits the formation of ubiquitin-positive protein aggregates in liver cells and neurons, suggesting that p62 promotes the degradation of selective protein cargo [152]. Previous studies have found that the 349-DPSTGE-354 motif in the Keap1-interacting region domain of p62, which resembles the Keap1-interacting ETGE motif in the Neh2 domain of Nrf2, accounts for the direct interaction between p62 and Keap1 [153]. The interactions between p62 and Keap1 isolates Keap1 in autophagosomes and disrupts Nrf2 ubiquitination degradation, thus activating the Nrf2 signaling pathway. Keap1 can be degraded by autophagy as a substrate regulated by p62. Therefore, p62 can change the half-life of Keap1. The overexpression of *p62* can significantly reduce the half-life of Keap1, while the knockdown of *p62* can prolong it [154]. In addition, Sestrin2 binds to p62, Rbx1, and Keap1 to form a protein polymer that promotes the autophagic degradation of Keap1 [155]. Other studies have found that p62 interacts with the Nrf2 binding site Keap1, so the excessive production or lack of p62 in autophagy can compete for the interaction between Nrf2 and Keap1, affecting the stability of Nrf2 and the transcriptional activation of the Nrf2 target genes [151]. In terms of structure and function, p62 contains a STGE motif, which can bind to the Kelch domain of Keap1 in the dissociated conformation state of the Nrf2-Keap1 complex [156]. The phosphorylation of p62 S351 in the STGE motif significantly increases the binding affinity between p62 and Keap1, leading to the depolymerization of Nrf2 and Keap1, resulting in the long-term accumulation of Nrf2 and increased transcription of Nrf2 target genes. Bae et al. found that Sestrins1 and Sestrins2 interact with Keap1, p62, and ubiquitin ligase Rbx1 and that the antioxidant effect of sestrins is activated by the Nrf2-mediated degradation of Keap1 by p62-dependent autophagy [155]. Furthermore, p62 regulates transcription factor Nrf2 by acting on the antioxidant proteins and detoxification enzymes of Nrf2.

The pathological process of oxidative stress can occur in a variety of diseases, and both autophagy and the Nrf2–Keap1 signaling pathway can be activated to participate in the pathological process. Various oxidative stress factors can induce *p62* gene expression, which is regulated by Nrf2, with the p62 protein promoting Nrf2 activation. Jain et al. confirmed that *p62* is the target gene for transcription factor Nrf2 and forms a positive feedback loop by inducing ARE-driven gene transcription [157]. Liao et al. found that *Nrf2* knockdown significantly reduces the expression of p62, and *p62* knockdown significantly reduces the expression of Nrf2, HO-1, and NQO1. Further studies have found that *p62* knockdown can increase the levels of apoptosis and ROS, suggesting that p62 can play an anti-oxidative stress role, and that this effect may be realized by regulating the Nrf2-Keap1 signaling pathway [158]. Kong et al. also found that the Nrf2-Keap1-p62 signaling pathway plays a significant role in resistance to oxidative stress injury [159]. Yang et al. found that phosphorylated p62 activation enhances Nrf2 nuclear translocation and reduces ROS production and mitochondrial dysfunction [160]. Chen et al. found that the crosstalk between oxidative stress and autophagy activity affects autophagy flux and inhibits downstream autophagy activity, and that the Nrf2-Keap-p62 signaling pathway is involved in interfering with normal cell functions [161]. In addition to p62, the autophagy adaptor protein NDP52 has been confirmed to participate in the crosstalk between the Nrf2 pathway and autophagy. Chulman et al. used *Nrf2* knockout rats and observed that the activation of the Nrf2 signaling pathway can reduce the level of neuronal phosphorylated tau by inducing the NDP52, which is strongly induced by Nrf2 and contains three ARE elements in its promoter region. The overexpression of this protein promotes the clearance of phosphorylated tau in the presence of autophagy-stimulating factors [162]. The crosstalk between Nrf2 and autophagy is summarized in Figure 4.

## 5. Crosstalk between Nrf2 and Autophagy in Ischemic Cerebrovascular Disease

After cerebral ischemia occurs, the blood supply to cerebral tissues in the ischemic area decreases, and various pathophysiological changes occur, including oxidative stress activation and autophagy hyperactivation [8]. Several studies have shown that oxidative stress is the main cause of nerve cell loss and tissue damage caused by ischemic brain injury [10,163]. In addition, overactivated autophagy causes neuronal damage in ischemic brain tissues. Therefore, we inferred that oxidative stress and overactivated autophagy may interact and aggravate brain tissue damage after cerebral ischemia. Studies have been conducted on the crosstalk between autophagy and oxidative stress in ischemic cerebrovascular disease. Studies on drugs for the treatment of ischemic stroke are abundant, but few studies have achieved clinical transformation. At present, rtPA thrombolysis and mechanical thrombectomy are still the main therapies used in clinical applications, but they pose side effects, such as hemorrhage transformation and BBB damage. The difficulty in the treatment of ischemic stroke mainly lies in the diversity of pathophysiological changes and the instability of disease progression. Li et al. studied the changes in autophagy activity and Nrf2 signaling pathway expression after cerebral ischemia and found that autophagy was over-activated, and the Nrf2 signaling pathway was inhibited after cerebral ischemia. The administration of estrogen (EST) after cerebral ischemia can largely restore Nrf2 expression and inhibit overactivated autophagy. However, this study did not explore the relationship between overactivated autophagy and the inhibited Nrf2 signaling pathway, nor did it thoroughly study the mechanism by which EST regulates autophagy and Nrf2 expression [164]. Amin et al. found that thymoquninone (TQ) could improve ischemic brain injury. The protective mechanism of TQ is to reduce inflammation and oxidative stress in ischemic brain tissue by activating the Nrf2/HO-1 signaling pathway and inhibiting over-activated autophagy and apoptosis. Nevertheless, this study did not reveal the mechanism by which the Nrf2 signaling pathway affects autophagy [165]. Cerebral I/R injury activates ER stress; using a cerebral I/R rat model, Wang et al. found that the expression of autophagy-related proteins, ATG12-ATG5 and LC3-PE, and the expression of *p62* mRNA were increased 1 h after cerebral ischemia and 24 h after cerebral reperfusion. In addition, the expression of Nrf2 downstream target antioxidant genes NQO1 and HO-1 increased, indicating the activation of autophagy and Nrf2 pathways. However, contrary results were observed 3 h after cerebral ischemia and 24 h after reperfusion, suggesting that autophagy and the Nrf2 pathway were inhibited. Consequently, they speculated that p62 could regulate ER stress after cerebral I/R injury, possibly through the regulation of the Nrf2-Keap1-ARE signaling pathway. The *p62* promoter contains ARE, and Nrf2 is the main protein that regulates ARE expression. Hence, they speculated that the Nrf2 signaling pathway may be involved in *p62* mRNA expression. However, they only made speculations based on the experimental results and previous mechanistic studies for other disease models and did not verify this in an animal model of cerebral ischemia. Recent studies have explored the crosstalk between Nrf2 and autophagy following cerebral ischemia. For instance, Liu et al. investigated the effects of TP53-induced glycolysis and apoptosis regulator (TIGAR) on cerebral ischemia and discovered that TIGAR can inhibit oxidative stress in animal models of cerebral ischemia and SH-SY5Y cells treated with OGD. TIGAR can induce autophagy at the cellular level, which is necessary for its antioxidant and neuroprotective effects. They further explored the related mechanisms, and found that TIGAR can reduce the expression of Keap1 in OGD-treated neurons, leading to an increase in Nrf2 expression and promoting nuclear translocation. Autophagy inhibitors can inhibit the nuclear translocation of Nrf2 and reduce the expression of Nrf2 target genes, suggesting that autophagy plays antioxidant and neuroprotective roles by activating Nrf2 [166]. Qi et al. found that (±)-5-bromo-2-(5-fluoro-1-hydroxyamyl) benzoate (BFB) promotes the nuclear translocation of Nrf2 by increasing related factors of the p62-Keap1-Nrf2 signaling pathway and confronts the oxidative stress injury induced by hydrogen peroxide in PC12 cells [167]. Crosstalk between Nrf2 and autophagy is not only realized by the interaction between p62 and Keap1. Cai et al. found that Isoquercitrin (ISO) promoted Nrf2 nuclear translocation and increased Nrf2 transcription in OGD-treated SH-SY5Y cells. Bioinformatic analysis identified that Aldolase C (ALDOC) is a potential target, and increased Nrf2 expression promoted ALDOC expression and enhanced autophagy, increasing the survival rate of SH-SY5Y cells [168]. Therefore, ALDOC is a key molecule linking the Nrf2-autophagy pathway.

Above are important insights for the interaction between Nrf2 and autophagy, but more attention is needed to clarify the detailed mechanisms.

## 6. Conclusions

Several clinical trials have proven that single-target drugs are ineffective in the treatment of cerebrovascular diseases and their complications, and a variety of drugs with different targets are used in clinical treatment [169]. The treatment of cerebrovascular diseases is currently a major challenge for clinicians because of the narrow time window for treatment after stroke onset and the existence of the BBB, which makes the transition of certain drugs difficult. Therefore, it is important to explore new treatments that target various pathophysiological targets of cerebrovascular diseases. The Nrf2 signaling pathway and excessive activation of autophagy in cerebral ischemia after damage to neurons and brain tissue are both vital factors, although there are few clear mechanisms for the interaction between these two pathophysiological changes in cerebrovascular diseases. Consequently, it is necessary to pay attention to this aspect and carry out in-depth and detailed research to explore new research directions for developing multi-target treatments for cerebrovascular diseases.

## Figures and Tables

**Figure 1 antioxidants-11-01747-f001:**
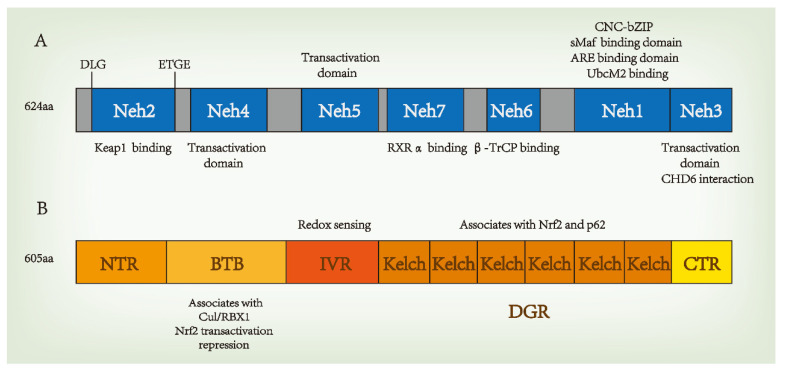
Domain structures of nuclear factor erythroid 2-related factor 2 (Nrf2) (**A**) and Kelch-like-ECH-associated protein 1 (Keap1) (**B**). β-TrCP, a β-transducin repeat-containing protein. CNC, cap‘n’collar; bZIP, basic region leucine zipper; sMafs, musculoaponeurotic fibrosarcoma protein; Cul3, Cullin 3; RXRα, retinoic X receptor alpha.

**Figure 2 antioxidants-11-01747-f002:**
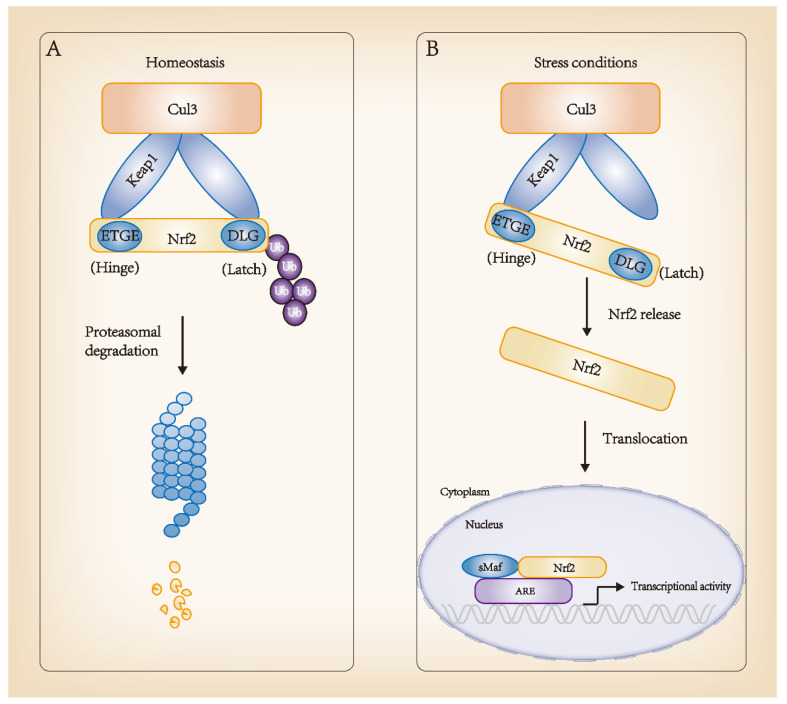
Regulation of Nrf2 molecular mechanisms. (**A**) Homeostasis: high affinity ETGE and low affinity DLG sites of Nrf2 bind to its inhibitor, Keap1, in the cytoplasm and interact with Cul3-Rbx1 E3 ubiquitin ligase, consistently leading to Nrf2 ubiquitination and proteasomal degradation. (**B**) Stress conditions: Keap1 changes its conformation and dissociates from Nrf2. Nrf2 translocates into the nucleus and forms a dimer with sMaf. The complex binds to ARE sequences, promoting the transcription of target genes, such as detoxification genes. ARE, antioxidant responsive element.

**Figure 3 antioxidants-11-01747-f003:**
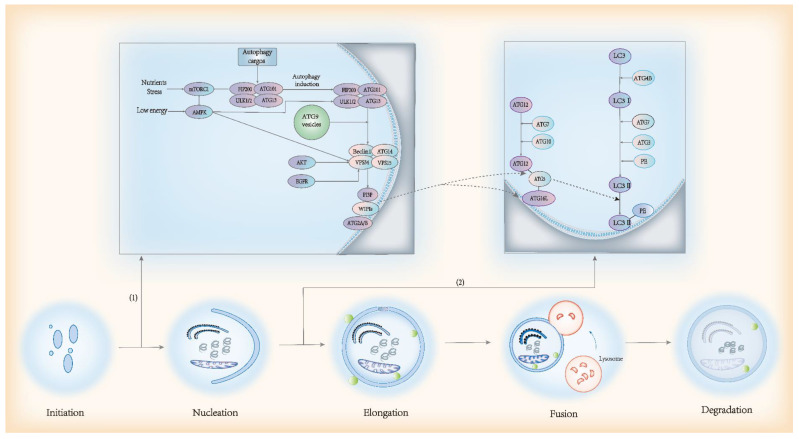
The classical process of autophagy. (**1**) Two types of activation signals (mTORC1 and AMPK activation signals and autophagic cargo activation signals) and proteasome complexes involved in the initial stage of autophagy. (**2**) Two ubiquitin-like conjugation systems involved in the elongation stage of autophagy.

**Figure 4 antioxidants-11-01747-f004:**
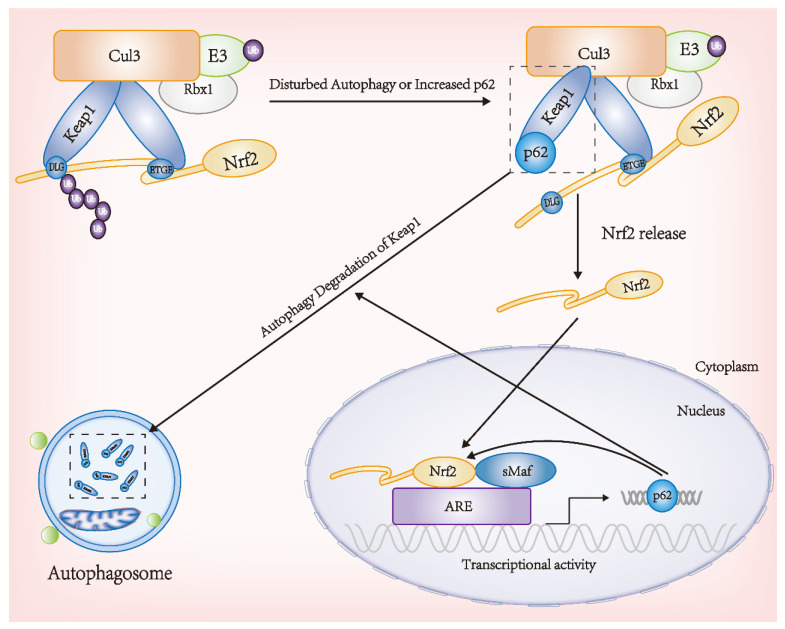
Crosstalk between Nrf2 and autophagy.

**Table 1 antioxidants-11-01747-t001:** Canonical and non-canonical function of autophagy-related protein.

Autophagy Process	Protein	Autophagic Role	Non-Autophagic Role(s)	References
initiation	FIP200	component of ULK complex (possibly scaffolding function)	pathogen control	[63,74]
	ULK1	serine/threonine kinase; initiates autophagy by phosphorylating components of the autophagy machinery	cytokine secretion, ER-to-GA anterograde transport, vesicular trafficking	[63,75,76,77]
	ULK2	initiates autophagy by phosphorylating components of the autophagy machinery	ER-to-GA anterograde transport, vesicular trafficking	[76,77]
	UVRAG	a binding partner and promoter of the BECN1/Beclin 1-associated lipid kinase PIK3C3/Vps34, autophagosome biogenesis	cell proliferation, centrosome functions, cytokinesis, DNA repair, endocytosis, GA-to-ER retrograde transport, LAP, melanogenesis, anti-inflammatory, tumor suppression	[78,79,80,81,82,83,84,85,86]
	ATG9	delivery of membrane material to the phagophore	ADCD, phagocytosis	[63,78,87]
	ATG14	PI3KC3–C1 targeting the PAS and expanding phagophore	ADCD	[63,88]
nucleation	ATG13	adaptor mediating the interaction between ULK1 and FIP200, enhances ULK1 kinase activity	ADCD, pathogen control	[63,74,89]
	VPS15	ULK-dependent phosphorylation	cytokinesis, endocytosis, mitochondrial metabolism	[82,90,91]
	VPS34	catalytic component of PI3KC3–C1, generates PI3P in the phagophore, and stabilizes the ULK complex	ADCD, cytokinesis, endocytosis, GA-to-ER retrograde transport, LAP, vesicular trafficking	[63,75,82,84,92]
	AMBRA1	downstream target of mTOR	ADCD, tumor suppression, cell proliferation	[93,94,95]
	BECN1	autophagosomes formation, extension, and maturation	ADCD, LAP, centrosome functions, cytokinesis, endocytosis, vision cycle, tumor progression	[79,82,84,88,96,97]
	BIF-1	colocalization with Atg5 and LC3, autophagosome formation	cytokinesis, endocytosis, tumor progress or suppression	[82,98,99]
	RUBCN	the class III PtdIns3K and HOPS complexes engagement	LAP, vision cycle	[84,96,100]
elongation	ATG3	E2-like enzyme, conjugation of activated ATG8s to membranal PE	cell proliferation, exosome secretion, LAP	[63,84,101,102]
	ATG4B	cysteine protease that processes pro-ATG8s, deconjugation of lipidated LC3 and ATG8s	granule exocytosis, LAP	[63,84,103]
	ATG5	E3-like complex that couples ATG8s to PE	ADCD, cell proliferation, exosome secretion, granule exocytosis, immunological memory, LAP, non-canonical protein secretion, pathogen control, vision cycle	[63,84,96,103,104,105,106,107,108,109]
	ATG7	E1-like enzyme, activation of ATG8, conjugation of ATG12 to ATG5	ADCD, cell proliferation, cytokine secretion, exosome secretion, granule exocytosis, immunological memory, LAP, pathogen control, PRR signaling	[63,84,92,103,106,107,110,111,112,113]
	ATG12	E3-like complex that couples ATG8s to PE	ADCD, exosome secretion, LAP, pathogen control	[63,84,102,104,114]
	ATG16L1	E3-like complex that couples ATG8s to PE	exosome secretion, LAP, phagocytosis, pathogen control, PRR signaling	[63,84,104,113,115,116]
cargo selection	LC3	interaction with autophagy receptors, phagophore expansion, and sealing	bacterial replication, cytokine secretion, granule exocytosis, LAP, pathogen control, vision cycle, viral replication, viral release	[63,84,96,106,111,113,117,118,119]
	NDP52	autophagy receptor	pathogen control	[63,112]
	p62	autophagy receptor	ADCD, pathogen control	[63,92,112]
fusion	RAB7A	correct targeting of ATG9a-bearing vesicles	endocytosis, exosome secretion, granule exocytosis, non-canonical protein secretion	[103,113,120,121,122]

ADCD, autophagy-related cell death; GA, Golgi apparatus; ER, endoplasmic reticulum; LAP, LC3-associated phagocytosis; PRR, pattern recognition receptor; LC3, microtubule-associated protein light chain 3; PtdIns3K, phosphatidylinositol 3-kinase; PE, phosphatidylethanolamine.

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
