# Peer review of "Nrf2 Pathway and Autophagy Crosstalk: New Insights into Therapeutic Strategies for Ischemic Cerebral Vascular Diseases"

_antioxidants, 2022, doi:10.3390/antiox11091747_

Round 1

Reviewer 1 Report

The present manuscript is a review article by Hu et al focusing on the Nrf2 pathway in cerebral vascular disorders. Overall the manuscript is comprehensive and well organized, the writing is mostly clear, the diagrams are informative and the content is good. I have 2 minor comments:

1 - The authors do a detailed description of the mechanisms regulating autophagy and their involvement in cerebral I/R, but little is explored of other cerebral pathologies that have a strong vascular component, such as cerebral amyloid angiopathy, cerebral small vessel disease and mixed dementias. The authors should include a brief discussion of the role of Nrf2 and autophagy in these diseases;

2 - The manuscript will benefit from a review of English.  

Author Response

  1. The authors do a detailed description of the mechanisms regulating autophagy and their involvement in cerebral I/R, but little is explored of other cerebral pathologies that have a strong vascular component, such as cerebral amyloid angiopathy, cerebral small vessel disease and mixed dementias. The authors should include a brief discussion of the role of Nrf2 and autophagy in these diseases;

Response:

  Thank you for your careful review.I have supplemented the information in the revised manuscript.

     Cerebral autosomal dominant arteriopathy with subcortical infarcts and leukoencephalopathy (CADASIL) is the most common form of hereditary brain small vessel disease characterized by NOTCH3 gene mutations. These patients were found to have SQSTM1 gene mutations, and the damaged NOTCH3 and p62 encode key proteins in the autophagic lysosomal pathway[1]. In addition, autophagic lysosomal pathway dysfunction was found in human CADASIL vascular smooth muscle cells[2]. Therefore, autophagy is an important pathological event of CADASIL. Damaged autophagy makes the clearance of dysfunctional mitochondria impaired, a large amount of mitochondrial DNA released into the cytosol to induce activation of caspase-1, and subsequently induced production of IL-1β, thus forming a low-grade sterile long-term inflammatory reaction[3]. In addition, aging diabetic rats are prone to cerebral small vessel disease accompanied by cognitive impairment, which is closely related to Beclin1-regulated autophagy in the hippocampus[4]. The important pathological changes of vascular dementia are cerebral microvascular injury and blood-brain barrier disruption. Autophagy plays a key role in maintaining vascular integrity, neuronal homeostasis and maintaining the integrity of the blood-brain barrier. After vascular dementia occurs, abnormal proteins, lipids and plaques are deposited in the brain. Regulated autophagy can inhibit the pathological process and maintain normal vascular physiology[5]. Abnormally activated autophagy reduces the clearance of protein aggregates and participates in the occurrence of pathological events. Cerebral amyloid angiopathy is an important cause of vascular dementia. The deposition of amyloid plaques in cerebral arteries is the main pathological event of cerebral amyloidosis. Abnormally activated autophagy promotes the formation of cerebral artery amyloid plaques, reduces the removal of protein aggregates, and increases the formation of vascular wall collapse[6].

    The early pathological change of cerebral small vessel disease is endothelial cell injury, and oxidative stress is the key factor of endothelial cell injury. Enhancing Nrf2 expression can significantly improve endothelial cell injury by reducing oxidative stress response[7]. Mixed dementia presents with mutually reinforcing neurodegeneration and vasculopathy, with all the features of Alzheimer's disease and vascular dementia. Enhancement of cAMP and/or cGMP signaling has been found to exert beneficial effects through activation of Nrf2[8].

  1. Almeida, M. R.;Silva, A. R.;Elias, I.;Fernandes, C.;Machado, R.;Galego, O.;Santo, G. C. SQSTM1 gene as a potential genetic modifier of CADASIL phenotype. J Neurol 2021, 268, 1453-1460, doi:10.1007/s00415-020-10308-5.
  2. Hanemaaijer, E. S.;Panahi, M.;Swaddiwudhipong, N.;Tikka, S.;Winblad, B.;Viitanen, M.;Piras, A.;Behbahani, H. Autophagy-lysosomal defect in human CADASIL vascular smooth muscle cells. Eur J Cell Biol 2018, 97, 557-567, doi:10.1016/j.ejcb.2018.10.001.
  3. Li, T.;Huang, Y.;Cai, W.;Chen, X.;Men, X.;Lu, T.;Wu, A.;Lu, Z. Age-related cerebral small vessel disease and inflammaging. Cell Death Dis 2020, 11, 932, doi:10.1038/s41419-020-03137-x.
  4. Guan, Z. F.;Zhou, X. L.;Zhang, X. M.;Zhang, Y.;Wang, Y. M.;Guo, Q. L.;Ji, G.;Wu, G. F.;Wang, N. N.;Yang, H.;et al. Beclin-1- mediated autophagy may be involved in the elderly cognitive and affective disorders in streptozotocin-induced diabetic mice. Transl Neurodegener 2016, 5, 22, doi:10.1186/s40035-016-0070-4.
  5. Wang, X. X.;Zhang, B.;Xia, R.;Jia, Q. Y. Inflammation, apoptosis and autophagy as critical players in vascular dementia. Eur Rev Med Pharmacol Sci 2020, 24, 9601-9614, doi:10.26355/eurrev_202009_23048.
  6. Berlit, P.;Keyvani, K.;Kramer, M.;Weber, R. [Cerebral amyloid angiopathy and dementia]. Nervenarzt 2015, 86, 1248-1254, doi:10.1007/s00115-015-4407-5.
  7. Li, M. T.;Ke, J.;Guo, S. F.;Wu, Y.;Bian, Y. F.;Shan, L. L.;Liu, Q. Y.;Huo, Y. J.;Guo, C.;Liu, M. Y.;et al. The Protective Effect of Quercetin on Endothelial Cells Injured by Hypoxia and Reoxygenation. Front Pharmacol 2021, 12, 732874, doi:10.3389/fphar.2021.732874.
  8. Sanders, O.;Rajagopal, L. Phosphodiesterase Inhibitors for Alzheimer's Disease: A Systematic Review of Clinical Trials and Epidemiology with a Mechanistic Rationale. J Alzheimers Dis Rep 2020, 4, 185-215, doi:10.3233/ADR-200191.

  I have made corresponding modifications and improvements in the manuscript, and retained the revision traces. The modification position is in the “2.2 Nrf2-involved in ischemic cerebrovascular diseases” and “3.3 Autophagy involved in ischemic cerebrovascular diseases” sections.

  1.  The manuscript will benefit from a review of English.  

Response:

   Thank you for your careful review, I have polished up the manuscript and retained the revision traces.

 The revised manuscript is submitted at the attachment.

Reviewer 2 Report

The manuscript entitled " Nrf2 pathway and autophagy crosstalk: New insights into therapeutic strategies for ischemic cerebral vascular diseases" describes the function of nuclear factor E2-related factor 2 (Nrf2), as well as the implication of this molecule ischemic cerebral vascular diseases. However, in this manuscript, some questions are still needed to address or fix.

Following are my comments:

11. In line 163, authors refer the “xenobiotic metabolic processes”, but they did not explain what are these processes or the pathway. A sentence about xenobiotic metabolic processes should be added.

22. In lines 175 and 176, authors described that “The Nrf2/Keap1 system plays a role in epigenetic regulation through DNA methylation, histone modification, and microRNAs.” Nrf2/Keap1 system leads to the induction of microRNAs expression or downregulation? What is the function of this microRNAs? Something is missing in this sentence.

33. In section 2, namely sections 2.2, 2.3 and 2.4, authors repeat the same information many times. These sections should be reorganized. On the other hand, the authors talk about the Nrf2 functions, then about diseases in which it is implicated and at the end they come back to the pathways linked to N Nrf2. This organization is very confusing.

44. The terms “Autophagic and non-autophagic functions of autophagy-related protein” should be change to canonical and non-canonical function of autophagy.

55. In table 1, the canonical functions of the ATG proteins should be presented first.

66. Authors should follow the guidelines for formatting Gene and Protein Names.

77. It is described that autophagy is involved in numerous diseases, thus, what it is described in the section “Diseases wherein autophagy is involved” is very reductive and should be reformulated.

88. The authors need to do some English editing. There are some errors throughout the text.

99. The figures are of poor quality. They should be improved.

Author Response

1.In line 163, authors refer the “xenobiotic metabolic processes”, but they did not explain what are these processes or the pathway. A sentence about xenobiotic metabolic processes should be added.

Response:

    Thank you for your careful review. Combined with the third review opinion you put forward, I revised 2.2,2.3,2.4 these three parts as one section “Nrf2 pathway-involved in ischemic cerebrovascular diseases”. Therefore, I removed the “Xenobiotic metabolic processes”, The following contents are not reflected in the revised manuscript.

   “Xenobiotic metabolic processes” is the process of removing exogenous compounds from the body at the participation of various metabolic enzymes. This process is completed by the regulation of stageâ… and stage â…¡ detoxification enzymes. The stageâ… detoxification enzymes are mainly responsible for the oxidation, reduction and hydrolysis of exogenous compounds to make them polar water-soluble metabolites. The main function of the stage â…¡ detoxification enzymes are to transfer endogenous hydrophilic groups so that the molecules can be easily excreted from the body. In the process of exogenous metabolism, the stageâ… and stage â…¡ detoxification enzymes may function simultaneously or separately due to the different molecular types.

2.In lines 175 and 176, authors described that “The Nrf2/Keap1 system plays a role in epigenetic regulation through DNA methylation, histone modification, and microRNAs.” Nrf2/Keap1 system leads to the induction of microRNAs expression or downregulation? What is the function of this microRNAs? Something is missing in this sentence.

Response:

    Thank you for your careful review. With the same situation as the first review opinion, I removed the “The Nrf2/Keap1 system plays a role in epigenetic regulation through DNA methylation, histone modification, and microRNAs”, The following contents are not reflected in the revised manuscript.

     Current studies predict that 85 microRNAs down-regulate Nrf2 translation by binding to Nrf2 mRNA, and 63 of them have positive and negative feedback loops with Nrf2[1]. Secondly, MicroRNA can also alter other proteins involved in Nrf2 expression, including Bach1 protein which competitively inhibit Nrf2 transcription, small Maf protein and DJ-1, Keap1 mRNA, and proteasome degradation protein. For example, studies have found that expression of miR200a in human breast cancer cell lines MDA-MB-231 and Hs578T can target Keap1 mRNA to enhance Nrf2 activation, while silencing miR200a can up-regulate Keap1 and inhibit Nrf2 activation[2]. It should be noted that different kinds of MicroRNA have different regulatory effects on Nrf2 pathway.

  1. Papp, D.;Lenti, K.;Modos, D.;Fazekas, D.;Dul, Z.;Turei, D.;Foldvari-Nagy, L.;Nussinov, R.;Csermely, P.;Korcsmaros, T. The NRF2-related interactome and regulome contain multifunctional proteins and fine-tuned autoregulatory loops. FEBS Lett 2012, 586, 1795-1802, doi:10.1016/j.febslet.2012.05.016.
  2. Eades, G.;Yang, M.;Yao, Y.;Zhang, Y.;Zhou, Q. miR-200a regulates Nrf2 activation by targeting Keap1 mRNA in breast cancer cells. J Biol Chem 2011, 286, 40725-40733, doi:10.1074/jbc.M111.275495.

  1. In section 2, namely sections 2.2, 2.3 and 2.4, authors repeat the same information many times. These sections should be reorganized. On the other hand, the authors talk about the Nrf2 functions, then about diseases in which it is implicated and at the end they come back to the pathways linked to Nrf2. This organization is very confusing.

Response:

    Thank you for your careful review, I have supplemented the information in the revised manuscript. Based on your valuable suggestion and that of another reviewer, I have integrated 2.2, 2.3,2.4 these three sections into one section, which echoes the title of this manuscript and reduces the misleading to readers.

    I have made corresponding modifications and improvements in the manuscript, and retained the revision traces. The modification position is in the “Nrf2 pathway-involved in ischemic cerebrovascular diseases” section.

  1. The terms “Autophagic and non-autophagic functions of autophagy-related protein” should be change to canonical and non-canonical function of autophagy.

Response:

   Thank you for your careful review. I have changed the terms “Autophagic and non-autophagic functions of autophagy-related protein” to “Canonical and non-canonical function of autophagy” in the revised manuscript. The modification position is in the title of the Table 1.

  1. In table 1, the canonical functions of the ATG proteins should be presented first.

Response:

   Thank you for your careful review. I have made corresponding modifications and improvements in the manuscript, and retained the revision traces.

  1. Authors should follow the guidelines for formatting Gene and Protein Names.

Response:

    Thank you for your careful review. I have made corresponding modifications and improvements in the manuscript, and retained the revision traces.

  1. It is described that autophagy is involved in numerous diseases, thus, what it is described in the section “Diseases wherein autophagy is involved” is very reductive and should be reformulated.

Response:

    Thank you very much for your valuable advice. Based on the third review comment you put forward above, in order to make the layout consistent, I changed the section of “Diseases wherein autophagy is involved” into the section “Autophagy involved in ischemic cerebrovascular diseases”, and rearranged the content. The reason why autophagy involved in the nervous system and tumors was only presented in the previous manuscript is that autophagy has been extensively studied in these diseases and have great value to deeply research. Besides, because the focus of this manuscript is ischemic cerebrovascular disease, so there is no too much introduction of other noncorrelation diseases, so as not to mislead the reader. In addition to the deletion of non-related diseases, other ischemic cerebrovascular diseases other than cerebral ischemia-reperfusion injury were added, including cerebral amyloid angiopathy, cerebral small vessel disease and vascular dementias.

I have made corresponding modifications and improvements in the manuscript, and retained the revision traces. The modification position is in the “Autophagy involved in ischemic cerebrovascular diseases” section and highlight in red.

  1. The authors need to do some English editing. There are some errors throughout the text.

Response:

    Thank you for your careful review. I have made corresponding modifications and improvements in the manuscript, and retained the revision traces.

  1.  The figures are of poor quality. They should be improved.

Response:

   Thank you for your careful review. I have made corresponding modifications and improvements in the Figure 1,3 and 4, and retained the revision traces.

Reviewer 3 Report

This review discusses the role of crosstalk after cerebrovascular disease in promoting oxidative stress and autophagy. Specifically, the action of Nrf-2 is elucidated. The authors offer an incredibly detailed description of the Nrf2 and autophagy pathways, discussing their roles in other diseases and then their roles in cerebrovascular disease specifically. Later, the interactive crosstalk between these two pathways is described. This review is well written and has informative and well-designed figures, but may be strengthened by the suggestions below:

Major suggestions:

  1. The presentation of the Nrf2 and autophagy pathways, while detailed and well written, requires additional in-depth discussion. Because these pathways of Nrf2 and autophagy are well described in the literature (Biochim Biophys Acta Mol Cell Res. 2018 May;1865(5):721-733; Autophagy. 2018;14(2):207-215), merely reciting this topic in the paper does not add much scientific value on what we already know about autophagy and Nrf2. Focusing on aspects of the pathways involved in autophagy and cross-talk as it relates to cerebrovascular diseases will strengthen the impact of the paper
  2. Are sections 2.2 and 2.4 necessary as distinct subheaders? Rather than have this discussion of function separate, which is further broken up by a discussion of Nrf2 in disease (section 2.3), it would be beneficial and more direct to simplify and combine these two sections to depict the functions of Nrf2 as a lead-in to the applications associated with disease states.

Minor suggestions:

  1. On line 105, double check the accuracy of the Neh domain in the sentence beginning with “Upon the 105 release of Neh1 from Keap1”.
  2. The authors state, on page 3 under the section ‘2.1 Structure of Nrf2,’ ‘The Neh5 domain also regulates Nrf2 cellular location, while Neh6 regulates the Keap1-dependent degradation of Nrf2 and represents a binding platform for the β-transducin repeat-containing protein (β-TrCP) [19].” A brief explanation of the relevance/function of β-TrCP would enhance the reader’s understanding of this structural feature of Nrf2.
  3. The neuroprotective role of Nrf2 in the development of Alzheimer’s disease is touched on in the beginning of section 2.3, but more details should be included when the topic is first introduced on page 6 line 211.
  4. Page 6 line 214 states that “Nrf2 is critical for the early progression of Parkinson’s disease”, and afterwards a study is cited as showing that Nrf2 plays a neuroprotective role in the development of Parkinson’s disease. A sentence or two explaining this proposed mechanism would help clarify the importance of Nrf2 in the context of Parkinson’s disease.
  5. Page 8 line 319 states “...we found that DMF significantly reduced…” however the study referenced seems to have been written by different authors, so the phrase “we found” would not be appropriate
  6. The authors state, on page 13 under the section ‘3.3 Diseases wherein autophagy is involved’ ‘Using genetically engineered mouse models of some cancers, such as lung cancer, autophagy has been found to inhibit the growth of benign tumors, but accelerate the growth of advanced tumors [166].’ Is there a proposed mechanism underlying this effect?
  7. It may be beneficial to shorten the sections on Nrf2 and autophagy pathways in other diseases. While these are beneficial to mention briefly, they distract the reader from the main purpose of the article.
  8. The table given is well detailed, but a more relevant table would include proteins involved in Nrf2 and autophagy and how they interact.
  9. Conclusion sentences for each paragraph and/or section would help guide the reader throughout the paper.
  10. Please check for grammatical and spelling errors. (i.e.) On page 1, “;Thus” should be lowercase. “Syudies” on page 7 should be “studies”. Too many spaces in line 464 and 510.
  11. Please ensure all abbreviations are described at first mention (i.e. rtPA)

Author Response

Major suggestions:

  1. The presentation of the Nrf2 and autophagy pathways, while detailed and well written, requires additional in-depth discussion. Because these pathways of Nrf2 and autophagy are well described in the literature (Biochim Biophys Acta Mol Cell Res. 2018 May;1865(5):721-733; Autophagy. 2018;14(2):207-215), merely reciting this topic in the paper does not add much scientific value on what we already know about autophagy and Nrf2. Focusing on aspects of the pathways involved in autophagy and cross-talk as it relates to cerebrovascular diseases will strengthen the impact of the paper.

Response:

    Thank you for your careful review. I have supplemented the information in the revised manuscript. I have made corresponding modifications and improvements in the manuscript, and retained the revision traces. The modification position is in the “Nrf2-involved in ischemic cerebrovascular diseases”, “Autophagy involved in ischemic cerebrovascular diseases” and “Crosstalk between Nrf2 and autophagy in ischemic cerebrovascular disease” sections.

  1. Are sections 2.2 and 2.4 necessary as distinct subheaders? Rather than have this discussion of function separate, which is further broken up by a discussion of Nrf2 in disease (section 2.3), it would be beneficial and more direct to simplify and combine these two sections to depict the functions of Nrf2 as a lead-in to the applications associated with disease states.

Response:

    Thank you for your careful review. I have supplemented the information in the revised manuscript. Combined with your suggestion and opinion of another reviewer, I revised 2.2,2.3,2.4 these three parts as one section “Nrf2-involved in ischemic cerebrovascular diseases” and rearranged previous content. In this section, I focused on depicting the functions of Nrf2 as a lead-in to the applications associated with ischemic cerebrovascular disease.

Minor suggestions:

  1. On line 105, double check the accuracy of the Neh domain in the sentence beginning with “Upon the 105 release of Neh1 from Keap1”.

Response:

   Thank you for your careful review. I found the reference 16 to proof the accuracy of the Neh domain in the sentence beginning with “Upon the release of Neh1 from Keap1”. I modified some of the nouns in the original sentence: signals-signal, migration-translocation.

4.The authors state, on page 3 under the section ‘2.1 Structure of Nrf2,’ ‘The Neh5 domain also regulates Nrf2 cellular location, while Neh6 regulates the Keap1-dependent degradation of Nrf2 and represents a binding platform for the β-transducin repeat-containing protein (β-TrCP) [19].” A brief explanation of the relevance/function of β-TrCP would enhance the reader’s understanding of this structural feature of Nrf2.

Response:

    Thank you for your careful review, I have supplemented the information in the revised manuscript. β-TrCP is a substrate adaptor for the S-phase kinase-associated protein 1 (SKP1)–Cul1–RING-box protein (Rbx1)/Roc1 ubiquitin ligase complex The Neh6 domain negatively regulates NRF2 through DSGIS and DSAPGS motifs. DSGIS motif increases the ability of β-TrCP to ubiquitinate Nrf2 and promotes its rapid conversion.

I have made corresponding modifications and improvements in the manuscript, and retained the revision traces. The modification position is in the “2.1 Structure of Nrf2, line 111 to 115” section.

  1. The neuroprotective role of Nrf2 in the development of Alzheimer’s disease is touched on in the beginning of section 2.3, but more details should be included when the topic is first introduced on page 6 line 211.

Response:

    Thank you very much for your valuable advice. The arrangement in section 2.3 in the original manuscript is to introduce Nrf2-related diseases against oxidative stress at first, and then introduce the neuroprotective effect of Nrf2 in neurodegenerative diseases including Alzheimer's disease and Parkinson's disease. However, at the beginning of this section, Alzheimer's disease is not mentioned. It is directly put forward in page6 line211. Combined with suggestions of other reviewers, in order to be more relevant to the topic of the article and avoid misleading the reader, I deleted Nrf2 and autophagy involved in other diseases that are irrelevant with ischemic cerebrovascular disease, and add some other than cerebral ischemia reperfusion injury of ischemic cerebrovascular diseases, such as cerebral amyloid disease, cerebral small vascular disease, and mixed dementia, etc. in the revised manuscript.

  I have made corresponding modifications and improvements in the manuscript, and retained the revision traces. The modification position is in the “2.2 Nrf2-involved in ischemic cerebrovascular diseases” section. However, this part is not reflected in the revised manuscript.

6.Page 6 line 214 states that “Nrf2 is critical for the early progression of Parkinson’s disease”, and afterwards a study is cited as showing that Nrf2 plays a neuroprotective role in the development of Parkinson’s disease. A sentence or two explaining this proposed mechanism would help clarify the importance of Nrf2 in the context of Parkinson’s disease.

Response:

   Thank you very much for your valuable advice. Oxidative stress and proteotoxic stress are two important pathophysiological events of Parkinson's disease. Loss of Nrf2 expression aggravates these two pathophysiological events. Impaired redox homeostasis and protein quality control lead to aggregation of pathogenic proteins, including α-synaptophysin(α-Syn), which play an important role in the progression of Parkinson's disease. Nrf2 loss increases phospho-α-Syn (hα-Syn) levels and enhances behavioral deficits in the hα-Syn overexpressing mice. α-Syn overexpressing mice develop higher oxidative stress levels in the absence of Nrf2. Cellular and molecular markers of neuroinflammation rise and autophagic changes in hα-Syn overexpressing mice with Nrf2 loss [1]. In conclusion, Nrf2 depletion is involved in the progression of PD through α-Syn and Nrf2 plays a neuroprotective role in PD.

Combined with suggestions of other reviewers, in order to be more relevant to the topic of the article and avoid misleading the reader, I deleted Nrf2 and autophagy involved in other diseases that are irrelevant with ischemic cerebrovascular disease, and add some other than cerebral ischemia reperfusion injury of ischemic cerebrovascular diseases, such as cerebral amyloid disease, cerebral small vascular disease, and mixed dementia, etc. in the revised manuscript. Therefore, this part is not reflected in the revised manuscript.

  1. Anandhan, A.;Nguyen, N.;Syal, A.;Dreher, L. A.;Dodson, M.;Zhang, D. D.;Madhavan, L. NRF2 Loss Accentuates Parkinsonian Pathology and Behavioral Dysfunction in Human alpha-Synuclein Overexpressing Mice. Aging Dis 2021, 12, 964-982, doi:10.14336/AD.2021.0511.

  1. Page 8 line 319 states “...we found that DMF significantly reduced…” however the study referenced seems to have been written by different authors, so the phrase “we found” would not be appropriate

Response:

   Thank you very much for your valuable advice. I have changed “we found” to “Yang et al. found” in page 7 line 305.

  1. The authors state, on page 13 under the section ‘3.3 Diseases wherein autophagy is involved’ ‘Using genetically engineered mouse models of some cancers, such as lung cancer, autophagy has been found to inhibit the growth of benign tumors, but accelerate the growth of advanced tumors [166].’ Is there a proposed mechanism underlying this effect?

Response:

    Thank you very much for your valuable advice.

Mechanisms involved in the accelerating the growth of advanced tumors: Autophagy confront mitigation of oxidative stress and remove damaged mitochondria to play a beneficial role in the growth of advanced tumors, thereby preventing activation of a senescence checkpoint that could limit tumorigenesis, and ensuring a continued supply of substrates for mitochondrial metabolism that supports the elevated metabolic and biosynthetic demands of a rapidly proliferating cancer cell.

Mechanisms involved in inhibiting the growth of benign tumors: Autophagy can combat oxidative stress, improve chronic tissue damage, and inhibit carcinogenic signaling by inhibiting the accumulation of damaged proteins and organelles.

     Based on the previous suggestions of reviewers, in order to make the layout consistent, I changed the section of “Diseases wherein autophagy is involved” into the section “Autophagy involved in ischemic cerebrovascular diseases”, and rearranged the content. Because the focus of this manuscript is ischemic cerebrovascular disease, so there is no too much introduction of other noncorrelation diseases, so as not to mislead the reader. In addition to the deletion of non-related diseases, other ischemic cerebrovascular diseases other than cerebral ischemia-reperfusion injury were added, including cerebral amyloid angiopathy, cerebral small vessel disease and vascular dementias. Therefore, the above mentioned is not reflected in the revision of the manuscript.

  1. It may be beneficial to shorten the sections on Nrf2 and autophagy pathways in other diseases. While these are beneficial to mention briefly, they distract the reader from the main purpose of the article.

Response:

      Thank you very much for your valuable advice, I have supplemented the information in the revised manuscript. Based on your valuable suggestion and that of another reviewer, I have integrated 2.2, 2.3,2.4 these three sections into one section, named “Nrf2-involved in ischemic cerebrovascular diseases”, which echoes the title of this manuscript and reduces the misleading to readers. Besides, I changed the section of “Diseases wherein autophagy is involved” into the section “Autophagy involved in ischemic cerebrovascular diseases”, and rearranged the content.

    I have made corresponding modifications and improvements in the manuscript, and retained the revision traces. The modification position is in the “2.2 Nrf2-involved in ischemic cerebrovascular diseases” and “3.3 Autophagy involved in ischemic cerebrovascular diseases” sections.

  1. The table given is well detailed, but a more relevant table would include proteins involved in Nrf2 and autophagy and how they interact.

Response:

     Thank you very much for your valuable advice. Figure 4 can display the proteins involved in Nrf2 and autophagy and how they interact more intuitively. I have revised Figure 4 to make above more concise.

  1. Conclusion sentences for each paragraph and/or section would help guide the reader throughout the paper.

Response:

    Thank you very much for your valuable advice. I have made corresponding modifications and improvements in the manuscript, and retained the revision traces.

  1. Please check for grammatical and spelling errors. (i.e.) On page 1, “;Thus” should be lowercase. “Syudies” on page 7 should be “studies”. Too many spaces in line 464 and 510.

Response:

    Thank you very much for your valuable advice. I have revised problems you put forward in the manuscript. I have made corresponding modifications and improvements in the manuscript, and retained the revision traces.

  1. Please ensure all abbreviations are described at first mention (i.e. rtPA)

Response:

     Thank you very much for your valuable advice. I have changed “mTORC1” to “mechanistic target of rapamycin complex 1 (mTORC1) in page 9 line 410, “Keap1” to “Kelch-like-ECH-associated protein 1(Keap1)” in page 2 line 94. I have made corresponding modifications and improvements in the manuscript, and retained the revision traces.

Round 2

Reviewer 1 Report

The authors have expanded their discussion of other cerebral vascular pathologies as requested, and improved the English grammar and writing to some extent. Some minor editing issues remains, but those do not preclude publication. Overall, I am quite pleased with the author's responsiveness.

Author Response

The authors have expanded their discussion of other cerebral vascular pathologies as requested, and improved the English grammar and writing to some extent. Some minor editing issues remains, but those do not preclude publication. Overall, I am quite pleased with the author's responsiveness.

Response:

     Thank you for your careful review. I have polished up the manuscript and retained the revision traces. The modification positions are in the page 6 line 235 to 242 and page 11 line 401 to 433.

Reviewer 3 Report

The authors have revised their manuscript according to my previous suggestions. This is now a much more solid paper!

Author Response

Thank you very much for your previous valuable suggestions.